# Regulation of Immunity in Breast Cancer

**DOI:** 10.3390/cancers11081080

**Published:** 2019-07-30

**Authors:** Chidalu A. Edechi, Nnamdi Ikeogu, Jude E. Uzonna, Yvonne Myal

**Affiliations:** 1Department of Pathology, Max Rady College of Medicine, University of Manitoba, Winnipeg, MB R3E 3P5, Canada; 2Department of Immunology, Max Rady College of Medicine, University of Manitoba, Winnipeg, MB R3E 0T5, Canada; 3Department of Physiology and Pathophysiology, Max Rady College of Medicine, University of Manitoba, Winnipeg, MB R3E 0J9, Canada

**Keywords:** breast cancer, immune response, immunotherapy, T and B cells, prolactin inducible protein

## Abstract

Breast cancer affects millions of women worldwide, leading to many deaths and significant economic burden. Although there are numerous treatment options available, the huge potentials of immunotherapy in the management of localized and metastatic breast cancer is currently being explored. However, there are significant gaps in understanding the complex interactions between the immune system and breast cancer. The immune system can be pro-tumorigenic and anti-tumorigenic depending on the cells involved and the conditions of the tumor microenvironment. In this review, we discuss current knowledge of breast cancer, including treatment options. We also give a brief overview of the immune system and comprehensively highlight the roles of different cells of the immune system in breast tumorigenesis, including recent research discoveries. Lastly, we discuss some immunotherapeutic strategies for the management of breast cancer.

## 1. Introduction

Worldwide, breast cancer is the second most frequently occurring cancer overall and the most common among women [1]. There were over 2 million new cases in 2018 which accounted for about 1 in 4 cancer cases in women, with the overall incidence rate just slightly below that of lung cancer [2]. In the last 40 years, the rate of breast cancer occurrence has been higher in women above 50 years of age while women below 50 years have had a decreased rate of survival [3]. Breast cancer can often lead to an enhanced economic burden on affected women and their families, as well as society. A US study estimated the annual cost of breast cancer care to be $16.5 billion [4].

Breast cancer can be broadly grouped into invasive lobular carcinoma and invasive ductal carcinoma [5]. However, there are many important differences within each group, and thus, a more accurate classification was warranted. Based on gene expression profiling, Sorlie et al. (2001) classified breast cancer into 5 subtypes, namely, luminal A and B tumors, human epidermal growth factor receptor 2 (HER2) over-expressing tumors, basal, and normal-like tumors [6,7]. This classification scheme has been shown to be useful in the prediction of patient outcomes and responses to chemotherapy. Other rare subtypes including molecular apocrine and claudin low subtypes have also been identified [8,9]. Furthermore, a study conducted in conjunction with the Molecular Taxonomy of Breast Cancer International Consortium (METABRIC) group led to the further classification of breast cancers into 10 subgroups/clusters based on a combination of gene expression profiles and copy number aberrations [10].

As with other cancers, several risk factors are known to significantly influence the odds of developing breast cancer. According to the Centre for Disease Control and Prevention (CDC), being female is a significant risk factor as less than 1% of breast cancers occur in males. Other risk factors include old age, genetic mutation (especially in the breast cancer (*BRCA*) genes), family history of breast cancer, and previous exposure to radiation therapy [11]. It has been suggested that women can decrease their risk of developing breast cancer by engaging in physical activity, limiting or avoiding alcohol intake, and controlling their weight [12]. Signs of breast cancer include: observing a mass in the breast, distortion of breast architecture, or asymmetry. Microcalcifications, identified by mammography, are also a common way by which breast cancer is detected. This may require further examination using a core needle biopsy, ultrasound, or magnetic resonance imaging (MRI) [13]. Like other carcinomas, breast cancer is staged (0–IV) based on the tumor node metastasis (TNM) system. This staging depends on tumor localization, metastasis, and disease severity with stage IV being the worst [14].

Common therapies for the management of breast cancer are surgery, radiotherapy, and systemic therapies, such as targeted therapy and chemotherapy. The treatment option used depends on the subtype or stage of the breast cancer. Since more than 60% of breast cancer patients have estrogen receptor positive (ER+) and progesterone (PR+) receptor positive tumors, endocrine therapy has been a game changer for many breast cancer patients [15]. A major breakthrough was the development of selective estrogen receptor modulators (SERMS)—for example, fulvestrant and tamoxifen—and aromatase inhibitors like letrozole. In addition, the introduction of targeted therapy with trastuzumab has been immensely beneficial for the 15% of patients with HER2+ tumors [16]. Different regimens have been developed for breast cancer chemotherapy, starting with the cyclophosphamide, methotrexate, and 5-fluorouracil (CMF) regimen, while epirubicin and cyclophosphamide, and subsequently paclitaxel (EC-paclitaxel), is the current standard of treatment. These regimens are used alone or in combination with other modes of treatment [17].

The potential of harnessing the body’s immune system for effective breast cancer treatment is gaining a lot of attention, with intensive research uncovering the complex interaction between breast cancer and the immune system. This has led to valuable insights that are being utilized in the development of promising immunotherapeutic strategies against breast cancer. This review summarizes the knowledge gained and recent insights into the regulation of immunity in breast cancer, with discussions on the roles of the innate and adaptive components of the immune system and ends by highlighting some immunotherapeutic strategies for breast cancer treatment.

### 1.1. Overview of the Immune System

The immune system is made up of many organ systems, cells, and the soluble bioactive molecules they produce, which recognize and defend the host against foreign proteins or antigens [18]. The immune system can be grouped into two main arms: the innate, which is immediate and non-specific; and the adaptive, which is specific and long-lasting [18]. The innate immune response is the first line of defense [19]. It can distinguish between “self” and “non-self” via toll-like receptors (TLRs) that can detect specific pathogen- or danger-associated molecular patterns (PAMPs or DAMPs) [18]. In addition, the innate immune system exerts its effects through proteins of the complement system as well as through cytokines [18]. Cytokines modulate different effects depending on which cells secrete them, where they are secreted, where their receptors are located, and the signaling pathways that are activated following their binding to the receptor [20]. The complement proteins, on the other hand, are activated by three major signaling pathways: classical, alternative, and lectin. Upon activation, these proteins mediate several effector functions, such as opsonization, recruiting other immune cells, and killing cells/pathogens by forming a membrane attack complex (MAC) for lysis [21]. Phagocytes and natural killer (NK) cells are important cellular components of the innate immune system. The phagocytes, such as monocytes, neutrophils, and macrophages are able to engulf cells expressing foreign or abnormal self-antigens and kill them in a process called phagocytosis [22]. On the other hand, natural killer cells secrete perforin and granzyme, which leads to the death of cells with abnormal major histocompatibility complex (MHC) class I or human leukocyte antigen (HLA) expression caused by pathogens or oncogenic mutations [22]. Mast cells, basophils, and eosinophils release inflammatory mediators, such as chemotactic leukotrienes, that attract immune cells to the inflammation or injury site [18]. There is yet another class of cells, referred to as NKT cells, that have features of NK cells and T cells [22].

The adaptive immune system has high specificity and leads to the development of immunological memory [18]. This form of immune response is not rapid because on encountering the appropriate antigens, naïve B and T lymphocytes need some time for differentiation and maturation into plasma cells (antibody-producing B cells) or effector T cells, respectively [18]. Based on the type of receptor, T cells are classified into 𝛼𝛽 T cells and 𝛾𝛿 T cells [23]. 𝛼𝛽 T cells like CD4^+^ T cells and CD8^+^ T cells require MHC-mediated antigen presentation [24,25], whereas 𝛾𝛿 T cells are able to recognize “non-self” antigens by pattern recognition [23]. Maturation of naive CD4^+^ T cells into effector cells requires co-stimulation between the T cell receptor and MHC class II (expressed on antigen-presenting cells like dendritic cells (DCs), B cells, and macrophages) [24,25]. CD4^+^ T cells are able to differentiate into different effector subsets such as T helper 1 (Th1), T helper 2 (Th2), T helper 17 (Th17), and regulatory T cells (Tregs), and this depends on the transcription factors and cytokines that are present [24]. These CD4^+^ T cell subsets produce and secrete immunomodulatory cytokines [24]. Unlike CD4^+^ T cells, which rely on MHC class II, naïve CD8^+^ T cells depend on MHC class I to mature into effector cytotoxic T lymphocytes [25]. The binding of activated CD8^+^ T cells via their receptors to the MHC class I-antigen complexes on target cells leads to the release of granzymes and perforin from the CD8^+^ T cells, which kills the target cells [25]. B cells secrete several antibodies which are highly specific [18] and act through various mechanisms, such as neutralization of antigens, induction of complement proteins, and antibody-dependent cell cytotoxicity (ADCC) [26]. Some immune system components, such as complement proteins, macrophages, and dendritic cells, serve as linkers between innate and adaptive immunity [27]. For example, macrophages and dendritic cells phagocytose pathogens (innate immunity) and present antigens to T cells to stimulate the adaptive immune response [19].

### 1.2. The Immune Response to Breast Cancer

From 1863, Rudolf Virchow suggested a connection between inflammation and subsequent cancer development [28]. Currently, many studies show the importance of the immune system in the initiation and progression of cancer [29]. In the initial stages of cancer development, cancerous cells are detected and eliminated by the innate immune system through a mechanism called immunosurveillance [29]. The abnormal growth of cancer cells can activate neighboring cells, which secrete tissue damage signals, such as interferon gamma (IFN-*γ*), that activate and recruit NK cells, the primary drivers of immunosurveillance [29]. It has been proposed that these processes control the appearance of cancers. Furthermore, in performing cancer immunosurveillance, the host immune system exerts pressure on a developing tumor, often eradicating cancerous cells before a tumor is established [30]. However, this same immune pressure is believed to influence tumor development and select for certain mutations, thereby creating an immune-evasive cancer.

The interaction between the immune system and cancer cells proceeds in three phases: elimination; equilibrium; and escape, which are referred to as the “three Es” of cancer immunoediting. During the ‘elimination’ phase, the immune system may succeed in destroying all tumor cells. If that does not happen, it may still be able to control tumor growth but not completely eradicate it. This phase is referred to as the ‘equilibrium’ phase. Finally, due to selection pressure from the immune system, some cancer cells develop enough resistance that they can escape the immune system, leading to a failure of immune-mediated cancer control [31]. This is referred to as the ‘escape’ phase. Cancers develop resistance by expressing reduced levels of MHC 1 and costimulatory molecules. They can produce factors that suppress the immune system, which enables them to avoid recognition by the immune system [32]. Interestingly, these immunosuppressive mechanisms may also be required for normal development and function of the mammary glands [33,34]. However, cancer cells can also utilize these same mechanisms to evade detection by the innate immune system and promote tumorigenesis. Thus, the tumor microenvironment becomes immunosuppressive and incapable of stimulating a potent adaptive immune response [35]. The current notion of the opposing nature of immune cells in tumors is that CD8^+^ T cells, CD4^+^ Th1 cells, NK cells, B cells, classically activated macrophages (M1), and mature dendritic cells contribute to tumor elimination. In contrast, CD4^+^ Th2 cells, regulatory B cells, CD4^+^ Tregs cells, myeloid-derived suppressor cells (MDSCs), and alternatively activated macrophages (M2) are pro-tumorigenic [36,37,38,39]. Generally, patients who have tumors with a Th2 cytokine profile have worse prognosis than patients with a Th1 or cytotoxic T lymphocyte (CTL) cytokine profile [40].

## 2. Role of the Innate Immune System in Breast Cancer Immunity

### 2.1. Innate Lymphoid Cells

The innate lymphoid cells include NK cells as well as other recently identified members such as group 1, 2, and 3 innate lymphoid cells (ILCs) [41]. ILCs have the morphology of lymphoid cells but do not possess somatically rearranged antigen receptors and lineage specific markers. ILCs are divided into groups 1, 2 and 3 ILCs, according to their cytokine and transcription factor expression [42]. The group 1 ILCs, which include NK cells, express the transcription factor T-bet. They respond to proinflammatory cytokines such as interleukins 12 (IL-12) and 18 (IL-18) and also produce type 1 cytokines such as IFN-*γ* and tumor necrosis factor alpha (TNF-α) [43]. The second group, group 2 ILCs (ILC2), need Gata3 and RORα (which are transcription factors) for their development and to produce their effector cytokines such as IL-5, 9, and 13 [44,45,46]. Group 2 ILCs enhance type 2 responses. Abnormal function of ILC2s can lead to allergic reaction, and has been implicated in the development of asthma, as well as lung and skin diseases [47]. Type 3 ILCs (ILC3) express the transcription factor RORγt, and produce IL-22 and IL-17A [48]. Members of this group include lymphoid tissue inducers (LTi) and natural cytotoxicity receptor (NCR) positive (NCR^+^) and NCR^−^ ILC3s [49]. ILC3s have been shown to contribute to the maintenance of homeostasis in the intestines. However, they have also been implicated in the pathogenesis of colitis [50]. In fact, based on effector functions, cytokine profiles and transcriptional activity, ILCs strongly behave like the CD4^+^ helper T cell subsets. ILC1s, ILC2s, and ILC3s are similar to Th1, Th2, and Th17 cells, respectively. NK cells can also be seen as the innate counterpart of CD8^+^ T cells [41].

Recent studies have reported an important role for innate lymphoid cells in breast tumorigenesis [51]. In a spontaneous breast cancer model, CD8^+^ T cells could not control early tumor growth, thereby suggesting that innate lymphocytes are critical for immunosurveillance during the early stages of tumorigenesis [51]. Also, in this model, IL-15-deficient animals (which lack ILC1 cells because IL-15 is important for their generation) showed accelerated tumor growth when compared to mice lacking CD8^+^ T cells. This suggests that ILC1s, but not CD8^+^ T cells, contribute to controlling early tumor growth. Because mice lacking Nfil3 (which do not have conventional NK cells) did not also show accelerated tumor growth in this model, the observation suggests that ILC1-like cells play an important role in mediating early tumor immunosurveillance [51]. Further studies are necessary to better understand the role of ILC1s in breast tumorigenesis.

As mentioned earlier, ILC2s enhance the pro-tumorigenic type 2 immune responses. For instance, IL-33 has been shown to stimulate ILC2s to induce type 2 responses. Administration of recombinant IL-33 in mice has been found to lead to increased breast tumor growth and metastasis [52]. IL-33 also stimulates ILC2s to secrete IL-13, which activates MDSCs and their production of transforming growth factor-β (TGF-β, an anti-inflammatory cytokine) [53]. Again, IL-13 is able to enhance the polarization of macrophages to the pro-tumorigenic M2 phenotype [54]. Therefore, it is plausible that ILC2s enhance tumor escape through IL-13-mediated stimulation of MDSCs and M2 macrophages, though direct evidence is lacking [54].

Although the role of ILC3s has been mostly investigated in the gut, there is enough evidence to indicate that the upstream mediator of ILC3s, IL-23, plays a role in carcinogenesis [41]. IL-23 stimulates ILC3s to produce IL-17A, which has been linked to angiogenesis as well as recruitment of MDSCs to the breast tumor site [55,56]. In contrast, ILC3-produced IL-22 has been reported to lead to reduced breast tumor growth [57]. Thus, ILC3s can have both pro- and anti-tumorigenic effects depending on the experimental model used.

### 2.2. Myeloid Cells

These are the most abundant hematopoietic cells and are critical for the normal function of the immune system. They can be divided into three groups: granulocytes, macrophages, and dendritic cells [58]. Myeloid cells contribute to the elimination of dying cells, protection of organisms from harmful pathogens, and tissue remodeling. Recently, their role in angiogenesis, invasion, and metastasis has been appreciated with evidence indicating that myeloid cells can be converted into immunosuppressive cells by the condition of the tumor microenvironment [59,60,61]. Tumor-infiltrating myeloid cells are comprised of granulocytes (basophils, eosinophils, and neutrophils), Tie2-expressing monocytes, DCs, tumor-associated macrophages, immature myeloid cells (IMCs), and MDSCs [62]. Studies have reported an expansion of infiltrating myeloid cells in breast cancer compared to normal breasts [62]. These cells have been found to exhibit granulocytic and immature myeloid cell phenotypes [63]. Recent studies have also shown that breast tumors produce the proinflammatory cytokine IL-1α, which then induces the production of thymic stromal lymphopoietin (TSLP) by tumor-infiltrating myeloid cells, especially neutrophils and monocytes [64]. TSLP then enhances tumor cell survival by inducing the expression of the anti-apoptotic protein Bcl2 [64]. Pinzon-Charry et al. have reported that in breast cancer patients, DC levels are decreased and they show more defects in function [65].

Macrophages are a group of myeloid cells that are closely related to DCs. In cancer, macrophages are usually influenced by molecules secreted by cancer cells to promote tumor growth and immune suppression. They can do this by producing IL-10, recruiting regulatory T cells, and inducing CD8^+^ T cell apoptosis [58]. In a mouse breast cancer model, mice have been shown to develop IL-4-producing Th2 cells, which polarized tumor-associated macrophages (TAMs) to the M2 phenotype. These TAMs then produced epidermal growth factor (EGF), which initiated invasion, migration, and metastasis of cancerous mammary epithelial cells [66].

Granulocytes are myeloid cells that possess cytoplasmic granules and a specific nuclear morphology. The most common type of granulocytes are neutrophils, which have been reported to promote angiogenesis in the tumor, and metastasis [58]. In a recent study, neutrophils were shown to suppress CD8^+^ T cell activity and promote metastasis in a spontaneous mouse breast cancer model [67]. In contrast, some other studies have shown that neutrophils produce reactive oxygen species (ROS) which contribute to the inhibition of metastasis [68]. Collectively, these reports suggest that like macrophages, neutrophils have contrasting roles in breast cancer.

MDSCs are immature CD11b^+^GR1^+^ cells that include precursors of granulocytes, macrophages, dendritic cells, and other myeloid cells [69]. They are produced in response to various breast cancer-derived cytokines and have been reported to inhibit T cell, NK cell, and DC functions while enhancing the activities of cancer immunoregulatory cells such as Th2 T cells, Tregs, and TAMs, thereby promoting breast tumorigenesis. MDSCs secrete cytokines such as IL-6 and IL-4 that act in an autocrine manner to enhance their expansion [69], and TGF-β, which has been reported to suppress antitumor immune responses [70]. There are two main types of MDSCs, namely, monocytic MDSCs and polymorphonuclear MDSCs (also known as granulocytic MDSCs). The mechanisms by which MDSCs modulate the immune system include promotion of Treg activity, depletion of nutrients needed by lymphocytes, generation of oxidative stress, and interference with lymphocyte trafficking [71,72,73]. Studies in humans suggest that MDSC levels in the peripheral blood positively correlate with disease burden and duration [71] and reduced levels may be correlated with improved efficacy of chemotherapy [69].

### 2.3. Effector Molecules of the Innate Immune System

Some components of the complement system have been shown to play either pro-tumorigenic or anti-tumorigenic roles. The finding of accumulation of the terminal pathway of complement (C5b-9 complement protein complex) on breast cancer samples has provided evidence of complement activation in breast cancer [74]. This also suggests that breast cancer cells may be potentially killed by complement activation through the membrane attack complex. Studies conducted in other cancer models have provided further evidence of the potential role of innate immune effector molecules in anti-tumor immunity. In melanoma, prostate, and ovarian cancers, it has been reported that the complement protein C1q promotes immunological tolerance and angiogenesis, as well as metastasis. In contrast, C1q has also been shown to be important for the efficacy of monoclonal antibody-based cancer immunotherapies. Another complement protein, C5a, promotes angiogenesis and has also been reported to be involved in the recruitment of MDSCs to the tumor site. On the other hand, C5a has been reported to be required for the differentiation of naïve CD4^+^T cells into Th1 cells and for the homing of T cells to the tumor site. C3a has been shown to enhance tumor progression and metastasis but also contributes to the infiltration of tumors by immune cells. The membrane attack complex has been shown to contribute to tumor killing [75,76]. Currently, there is evidence to suggest that host defense peptides, which have been studied for their role as antimicrobials, also possess anti-tumor activity [77]. These cationic host defense peptides are capable of electrostatically interacting with phosphatidyl serine, usually found on the outer membrane of cancer cells. This leads to the disruption of the plasma membrane (and in some cases, the mitochondrial membrane), resulting in apoptosis of cancer cells [77].

## 3. Role of the Adaptive Immune System in Breast Cancer Immunity

### 3.1. B Cells

B cells regulate the immune response in cancer through the production of antibodies and IL-10, and by interaction with other immune cells [78]. Tumor-infiltrating B cells have been reported to secrete antibodies against tumor antigens such as β-actin and ganglioside GD3 [79]. In a 4T1 breast cancer model, activated tumor-draining lymph node (TDLN) B cells were shown to secrete immunoglobulin G (IgG) in response to cancer cells and were also able to specifically mediate the killing of cancer cells *in vitro*. When these activated B cells alone were transferred *in vivo*, it led to inhibition of metastasis and induction of a T cell immune response against the tumor [80]. In addition, when IL-2 was administered along with TDLN B cells, there was increased production of IgG which mediated 4T1 cancer cell lysis in a CXCR4/CXCL12-dependent mechanism through Fas/FasL pathways and other mechanisms [80]. Conversely, regulatory B cells (Bregs) in the breast tumor microenvironment can secrete IL-10 and TGF-β, which enhance regulatory T cell activity that is known to inhibit antitumor immunity. Bregs also express PDL1, which suppresses proliferation and activation of tumor-specific T lymphocytes [81].

The prognostic function of tumor-infiltrating B cells, particularly in breast cancer, has been controversial, with different studies reporting positive, negative, or no prognostic value. A recent meta-analysis [82] has reported increased numbers of CD20^+^ B cells which were associated with better disease-free survival (DFS) and breast cancer specific survival (BCSS). In contrast, another study [72] showed that CD19^+^ B cell levels were positively associated with indicators of poor prognosis such as estrogen receptor/progesterone receptor negative ( ER-/PR-) tumors, metastasis to the lymph nodes, histological grade III tumors, and TNM stage T4 status. Furthermore, studies have reported a weak correlation between CD20^+^ B cell levels and histological grade, cyclin A, and Ki67 in breast cancer [83]. Altogether, these studies suggest that B cells have different prognostic values depending on the disease condition.

### 3.2. T Cells

Both 𝛼𝛽 (CD4^+^ and CD8^+^ T cells) and 𝛾𝛿 T cells are critical in antitumor immunity. The presence of T cells in tumors and their degree of infiltration are positive prognostic indicators when compared to non-immunogenic tumors. This has been shown to correlate with positive prognostic indicators such as lower histological grade of tumor, axillary lymph node negativity, reduced tumor size, and recurrence-free survival [84], thus indicating the central role of T lymphocytes in immune surveillance. However, the precise composition of tumor-infiltrating T cells varies and can greatly impact tumor progression and overall patient survival [61].

CD8^+^ T lymphocytes give rise to CTLs, which are the major effector cells against breast cancer [85]. These CTLs recognize specific antigens that are presented by cancer cells through MHC class 1, and then release perforin and granzymes, which kill cancer cells [86]. Naive CD4^+^ T cells, on the other hand, are able to differentiate into specific effector subtypes, including, but not limited to, classical T helper -1, -2, and -17 (Th1, Th2, and Th17), and regulatory T cells [24]. Th1 cells secrete IL-2, IFN-γ, and TNF-α, which promote macrophage cytolytic and antitumor activities. Th2 cells, on the other hand, secrete IL-4, IL-5, IL-6, IL-10, and IL-13, which induce anergy in T cells and enhance the activities of tumor-promoting macrophages [24]. In a recent *in vivo* study, IL-17A produced by Th17 cells caused tumor growth by changing the behavior and gene-expression profile of non-metastatic tumor cells [87], thus indicating a pro-tumorigenic role for IL-17A. In addition, elevated expression of IL-17A was correlated with worse prognosis and disease-free survival in patients with invasive ductal carcinoma (IDC) of the breast [87]. Tregs have been reported to promote tumor progression in breast cancer by suppressing the activities of CTLs and Th1 cells [88]. In addition, intra-tumoral FoxP3 expression (a marker for Tregs) is linearly associated with invasion, size, and vascularity of the tumor [88].

The roles of different immune cells in breast cancer are summarized in Figure 1.

In addition to cytokines like IL-2, IL-12, TNF-α, and IFN-γ, which have been reported to mediate the Th1 immune response [24], our group identified the prolactin inducible protein (PIP), an abundantly secreted protein in some human breast cancer cell lines [89,90], as an important immunoregulatory protein that regulates Th1 immune responses [91]. Interestingly, more than 90% of breast cancers express PIP to varying degrees and recent studies show that PIP expression is associated with better prognosis and patient response to chemotherapy [92,93]. The role of PIP in breast cancer is not known. However, we have found that in addition to a role in innate immunity [94], PIP also plays a role in the adaptive immune response. Specifically, we demonstrated that loss of PIP *in vivo* led to defective type 1 T helper cell (Th1) activity, which is important for antitumor immunity [95]. These observations suggest that PIP may inhibit breast tumorigenesis by enhancing antitumor immunity. As shown in Figure 2, we propose that PIP from breast cancer cells binds to its putative receptor on dendritic cells, leading to increased production of IL-12, which enhances CD4^+^ Th1 differentiation and IFN-γ production. IFN-γ can directly inhibit the proliferation of breast cancer cells. In addition, IFN-γ can activate the antitumorigenic M1 macrophages which phagocytose breast cancer cells. PIP expression has also been shown to enhance intracellular cytokine signaling in macrophages by activating mitogen-activated protein kinase (MAPK) and signal transducer and activator of transcription proteins (STAT) and inhibiting the expression of suppressors of cytokine signaling (SOCS) [96], all of which contribute to enhanced macrophage activity. Understanding the function of PIP in breast tumorigenesis is important because if confirmed to enhance immunity against breast cancer, it may be a viable option in immunotherapy.

## 4. Immunotherapy

The activation of the immune system for therapeutic benefits, which is termed immunotherapy, has been shown to be effective in treating several cancers, especially under experimental conditions. Immunotherapy relies on the fundamental discovery that cancer cells express specific tumor-associated antigens, some of which elicit measurable and sometimes protective humoral and/or cellular immune responses. The overall concept and key steps in the development of cancer immunotherapeutic strategies are focused at recruiting the host’s immune cells and tailoring their ability to identify and destroy tumor cells in an antigen-specific manner.

The ability to stimulate tumor immunogenicity by converting a low tumor-infiltrating lymphocyte (TIL) tumor to a high TIL tumor is generating much attention in cancer immunology. Changes in TIL levels and composition have been monitored in mouse models and samples from patients after chemotherapy [97,98]. This increased TIL infiltration is proposed to be stimulated by immunogenic cell death or apoptosis, which leads to the release of tumor neoantigens, thereby resulting in enhanced antigen uptake and presentation by dendritic cells [99,100]. These observations have led to the development of therapeutic strategies, where conventional anticancer chemotherapies are combined with a class of immunotherapeutic agents called checkpoint inhibitors [101].

### 4.1. Checkpoint Inhibition

The immune system has evolved to possess immune suppressive mechanisms like immune checkpoints which protect against autoimmunity by promoting tolerance to self-antigens. Tumors exploit these immunosuppressive mechanisms to weaken antitumor responses, thereby escaping detection and elimination by the immune system [102]. Checkpoint inhibitors are antibodies that bind to checkpoint molecules, leading to their deactivation. Ipilimumab (YervoyR, Bristol-Myers Squibb, New York, NY, USA) was the first immune checkpoint inhibitor to undergo clinical trials. It targets a checkpoint molecule known as cytotoxic T lymphocyte associated protein-4 (CTLA-4) and was approved for the management of melanoma [103]. CTLA-4 is an inhibitory receptor expressed on T cells which interacts with its ligand and suppresses T cell activation and hence immune response [102]. In breast cancer, anti-CTLA-4 treatment was used for the first time in a Phase I trial to treat patients with advanced ER+ breast cancer. In this study, anti-CTLA-4 immunotherapy was combined with tremelimumab and exemestane and an elevation in the levels of peripheral CD4^+^ and CD8^+^ T cells, which express inducible co-stimulatory molecules (ICOS), was observed, suggesting a role for anti-CTLA-4 inhibitors in breast cancer treatment [104]. However, there was no correlation between total lymphocyte counts or CD4^+^ and CD8^+^ T cell numbers and clinical response. The best response reported was durable disease for 12 weeks or greater in about 42% of patients. Although this treatment led to prolonged survival, there is a need to develop therapies with better efficacy, and clinical trials that include a larger sample size and a heterogenous population need to be conducted. In another study using ipilimumab (anti-CTLA4) in early stage breast cancer, the researchers observed that at a dose of 10 mg/kg, ipilimumab was tolerated and led to increase in antitumor Th1 response as well as ICOS levels. This suggests that anti-CTLA4 therapy with ipilimumab enhances antitumor immunity [105]. Expression of another checkpoint molecule, programmed death ligand (PD-L1), on breast tumors cells and stromal cells has been studied in depth via analysis of pathological specimens [106]. PD-L1 binds to its receptor, PD-1, which is found on the surface of T cells and causes inactivation or exhaustion of the T cells. Early phase clinical trials using anti-PD1 and anti-PD-L1 monotherapy have indeed shown therapeutic potential [101]. PDL1 levels in the tumor are determined by immunohistochemistry and expression levels of ≥1% are considered positive and are correlated with a better response to PD-1 and PD-L1 inhibitors. In one study where 32 TNBC patients were treated with pembrolizumab (an anti-PD-1 antibody), 4 patients displayed severe toxicity and 1 death was recorded. Out of the 27 patients remaining, the overall response rate was 18.5% [107]. In another Phase I trial, 168 patients with metastatic breast cancer were treated with avelumab, an anti-PDL1 antibody, for 2–50 weeks and monitored. Like in the previous study, there were mild to severe toxicities with 2 deaths recorded. The response rate was 3.0% for all breast tumors and 5.2% for triple negative breast tumors. In addition, tumors demonstrating PD-L1 positivity had a remarkably higher response rate (16.7%) compared to the general sample population (3.0%) [108].

### 4.2. Combination Therapy

Due to the relatively low response rate for checkpoint blockade monotherapy, it has become necessary to enhance responses by combining treatments that increase immunogenicity with those that relieve mechanisms of immune escape. For instance, several promising clinical trials are in progress where conventional anticancer treatments are combined with anti-PD-1/PD-L1 checkpoint blockade [101]. In addition, the combination of trastuzumab with checkpoint blockade is being considered for use in the treatment of HER2+ breast cancer as its administration has been shown to lead to improved therapeutic efficacy in an animal model with primary resistance to immunotherapy [109]. Checkpoint blockade may also have synergistic effects when combined with kinase inhibitors in targeted treatments. The inhibition of MEK (also known as mitogen-activated protein kinase kinase (MAPKK)) has been shown to be capable of relieving immune suppression in animal models of TNBC, which have an alteration in the Ras-MAPK pathway [110]. Therefore, MEK inhibitors may be combined with anti-PD-1/PD-L1 checkpoint blockade in TNBC patients showing similar features as the animal model [101].

### 4.3. Emerging Immunotherapies

While checkpoint inhibitors are used to revitalize exhausted immune responses that already exist, techniques such as adoptive cell transfer are used for expanding tumor-specific T cell populations. In a ground-breaking study conducted at the National Institutes of Health (NIH), a patient with metastatic breast cancer was screened for cancer specific mutations. The TILs specific for these mutations were isolated, expanded ex vivo, and adoptively transferred to the patient together with IL-2 and pembrolizumab (a checkpoint inhibitor). Interestingly, this treatment led to complete and durable disease regression in the patient [111]. Although only one patient was treated in that study, the results were encouraging and need to be conducted in a larger cohort to assess reproducibility and to optimize the treatment protocol. In another landmark study, chimeric antigen receptor (CAR)-T cell therapy targeting the tumor antigen MUC1 was shown to be effective against breast cancer in mouse models [112]. Clinical trials are currently ongoing using CAR-T cells that target antigens such as mesothelin and ROR1 in breast cancer [113]. Almost 60% (13 out of 22) of patients with advanced breast cancer had delayed disease progression when bispecific antibodies for HER2 and CD3 were combined with IL-2 and granulocyte monocyte colony stimulating factor (GM-CSF) in a Phase I clinical trial [114].

Vaccination is another emerging immunotherapy that has stimulated the interest of breast cancer researchers. Several cancer vaccines have been developed from different immunogenic sources such as DNA, RNA, tumor lysates, tumor antigenic peptides, and viruses. These vaccines can be combined with immunoadjuvants like GM-CSF. Reports from clinical trials with vaccines targeted against HER2+ breast cancer and other highly expressed antigens like human telomerase reverse transcriptase (hTERT) have been encouraging [115]. For instance, E75, a peptide derived from the HER2 protein, was used as a vaccine in combination with GM-CSF as adjuvant. In response to the vaccination therapy, 50% of the patients tested elicited an E75-specific T cell activity which destroyed the HER2+ breast cancer [116]. Furthermore, a Phase III clinical trial conducted using MUC1 for treating patients with ER+ stage 2 breast cancer showed that the MUC1-vaccine-treated group displayed a significantly reduced recurrence rate (12.5%) compared to the placebo control (60%) [117].

Current immunotherapies used against breast cancer are summarized in Figure 3.

## 5. Epigenetic Regulation of Immunity to Breast Cancer

Epigenetic changes are now known to also play a role in breast cancer development and immune regulation [118]. Epigenetic processes like histone modification, DNA methylation, and remodeling of the nucleosome are capable of activating or inhibiting gene expression [118] which may in turn activate or inhibit antitumor immune responses [119]. In addition, histone deacetylase inhibitors and DNA methyltransferase (which are epigenetic drugs) are able to activate antitumor immunity via mechanisms such as enhancing the expression of tumor-associated antigens, antigen processing and presentation, chemokines, and immune checkpoint inhibitors [120].

## 6. Conclusions/Perspectives

Breast cancer is the most common cancer among women with significant death rates and economic impact. Although the survival rates have greatly improved over the years with advances in research and therapy, there is still more to be done. Current treatment modalities include surgery, radiotherapy, chemotherapy, hormone therapy, and, more recently, immunotherapy. Accurate and reliable information including tumor stage, grade, and, more recently, subtype is now deemed necessary to decide the appropriate therapeutic approach for breast cancer patients. Early stage and low-grade breast cancers may be treated with surgery. For hormone receptor positive (ER+, PR+) cancers, hormonal therapy is still usually the treatment of choice. HER2+ breast cancer patients are treated with trastuzumab while TNBC patients are usually treated with chemotherapy. In addition, the genomic mutational status as identified by platforms such as Oncotype DX is also important as it provides information about potential genes that can be targeted with the right inhibitors or antibodies. For immunotherapy, it has been shown that PD-L1 expression levels in the tumor correlates with response to anti-PD-1 and anti-PD-L1 therapies. The degree of tumor-infiltrating lymphocytes and immunogenicity also contribute to response to immunotherapy. TNBCs and HER2+ breast cancers usually have higher percentages of TILs and are more immunogenic compared to hormone receptor positive breast cancers. Thus, they respond better to immunotherapy [121,122].

Ongoing studies are uncovering novel findings about the molecular mechanisms of the interaction between breast cancer and the immune system, which have in turn led to an improved understanding of the fundamental principles of this interaction. In addition, knowledge from these findings have indeed been applied in the development of immunotherapeutic strategies for breast cancer treatment. For instance, breast cancer was previously thought to be unamenable to immunotherapy due to its low immunogenicity. However, recent findings have reported some level of immunogenicity, which has been exploited in the development of immunotherapeutic agents against breast cancer, such as checkpoint inhibitors and vaccines. Again, studies now show that chemotherapy greatly enhances the immunogenicity of breast tumors, thereby paving the way for combination therapies where chemotherapeutic agents are used concurrently with checkpoint inhibitors to greatly enhance therapeutic efficacy. In addition, studies have shown that the immune system can have both pro-tumor and antitumor functions. Altogether, a better understanding of the role each subset of the immune system plays is warranted. This knowledge will be vital in improving the survival of breast cancer patients. Thus, strategies could be developed to either suppress the tumor promoting subset(s) of the immune system, enhance the antitumor subsets of the immune system, or utilize a combination of both strategies.

## Figures and Tables

**Figure 1 cancers-11-01080-f001:**
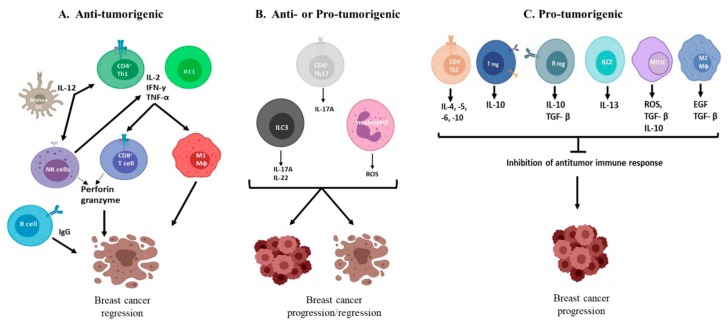
The role of different immune cells in breast cancer. **Panel A** shows anti-tumorigenic immune cells. Mature dendritic cells (DCs) present antigens to T cells and secrete IL-12 which enhances anti-tumorigenic CD4^+^ Th1 and natural Killer (NK) cell immune responses. CD4^+^ Th1 and ILC1 secrete IL-2, IFN-γ and TNF-α which stimulate anti-breast tumor immune activity by activating effector cells such as cytotoxic T cells (CD8^+^ T cells) and M1 macrophages (Mφ). NK cells also secrete IFN-γ and TNF-α. CD8^+^ T cells and NK cells secrete perforin and granzyme which directly kill breast cancer cells while B cells secrete IgG which has anti-tumor activity. **Panel B** shows immune cells that possess both anti- and pro-tumorigenic activities. CD4^+^ Th17 cells and ILC3s produce IL-17A which has been shown to be both anti- and pro-tumorigenic in breast cancer. ILC3s also secrete IL-22 which has been reported to reduce breast cancer growth. Neutrophils can suppress CD8^+^ T cell function and secrete reactive oxygen species (ROS) which kill breast cancer cells. **Panel C** shows pro-tumorigenic immune cells. CD4^+^ Th2 secrete IL-4,-5,-6 and -10 while ILC2s secrete IL-13. Regulatory T (Treg) and B (Breg) cells secrete IL-10 and TGF-β which suppress anti-tumor immune responses. M2 macrophages and myeloid derived suppressor cells (MDSCS) also secrete the TGF-β and other factors which dampen anti-tumor immune responses and stimulates tumor growth. IL (interleukin), Th1 (type 1 T-helper cells), ILC1 (group 1 innate lymphoid cells), IFN-γ (interferon-gamma), TNF-α (tumor necrosis factor-alpha), M1 macrophages (classically activated macrophages) IgG (immunoglobulin G), Th17 (type 17 T-helper cells ), ILC3 (group 3 innate lymphoid cells), Th2 (type 2 T-helper cells), ILC2 (group 2 innate lymphoid cells), TGF-β (transforming growth factor-beta). M2 macrophages (alternatively activated macrophages).

**Figure 2 cancers-11-01080-f002:**
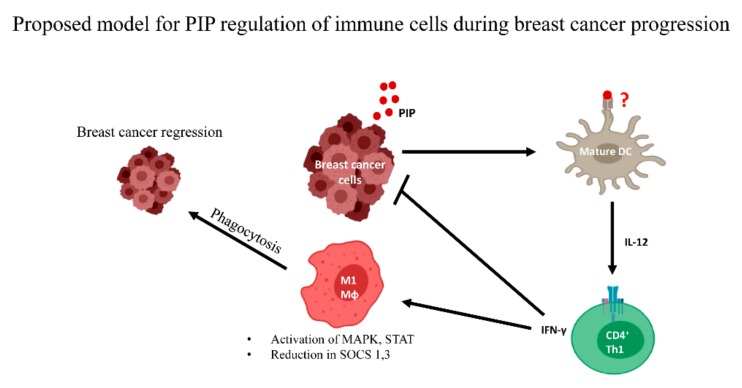
Proposed immune regulatory role for prolactin inducible protein (PIP) during breast cancer progression. During breast cancer development, it is proposed that PIP secreted by breast cancer cells binds to a receptor (unknown) on dendritic cells (DC) leading to enhanced IL-12 production which enhances the production of IFN-γ by CD4^+^ Th1 cells. IFN-γ then directly inhibit breast cancer progression. As well, it can act indirectly by enhancing intracellular signaling events through MAPK and STAT, resulting in the activation of M1 macrophages (Mφ) which phagocytose and destroy breast cancer cells. MAPK: mitogen activated protein kinase, STAT: signal transducer and activator of transcription proteins, SOCS: suppressors of cytokine signaling.

**Figure 3 cancers-11-01080-f003:**
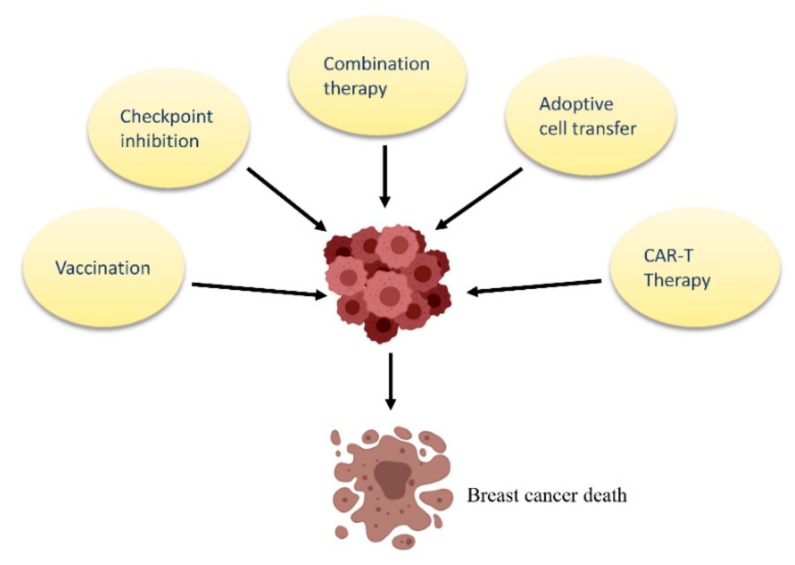
Current immunotherapeutic strategies against breast cancer. Clockwise. *Vaccination* involves the use of tumor antigens or other immunogenic molecules to stimulate anti-tumor immune response. *Checkpoint inhibition* is an immunotherapeutic approach which inhibits checkpoint molecules using antibodies, thereby enhancing anti-tumor immune response. *Combination therapy*: chemotherapeutic agents are combined with checkpoint inhibition to enhance anti-tumor immune response. *Adoptive cell transfer* of expanded tumor specific immune cells has shown potent anti-tumor activity. *Chimeric antigen receptor T cell (CAR-T) therapy* involves genetic modification of T cells to allow them to better recognize tumor antigens, leading to enhanced anti-tumor immune response.

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
