# Peer review of "Regulation of Immunity in Breast Cancer"

_cancers, 2019, doi:10.3390/cancers11081080_

Round 1
Reviewer 1 Report
Regulation of immunity during breast cancer
This review offers a good perspective of current views on tumor biology in general and breast cancer specifically. There are a few remarks for improvement
Figure 1 is very small
It is more informative to add response rates to check point inhibition therapy and other treatments
How much PD-L1 and on which cells is needed to treat with check point inhibitors?
The text on PIP does not belong in the perspectives, it should be in the immune system part or in the breast cancer specific immunity part.
In perspectives I would prefer indications how to decide which therapy to use, or which information will be needed to collect before we can make decisions on personalized treatment in breast cancer.
Author Response
We thank the reviewers for their constructive and insightful comments which have greatly enhanced the quality of our manuscript. We have addressed all the issues raised by the reviewers and have done additional literature search to strengthen the overall quality of the manuscript.
REVIEWER 1
Comment 1. Figure 1 is very small
Response: We agree with the reviewer and as suggested, we have increased the size of Figure 1 to enhance readability (page 8).
Comment 2. It is more informative to add response rates to check point inhibition therapy and other treatments.
Response: The authors agree with the reviewer and in the revised manuscript, we have added the response rates to different immunotherapeutic strategies under the sections “Checkpoint Inhibition” and “Emerging Immunotherapies” (pages 10 and 11 respectively).
Comment 3. How much PD-L1 and on which cells is needed to treat with check point inhibitors?
Response: PD-L1 expression level on the breast tumor cells is usually determined by immunohistochemistry and an expression level of ≥1% is considered PD-L1 positive. However, studies suggest that these PD-L1 positive tumors respond better to treatment with anti-PD1 and anti-PD-L1 checkpoint inhibitors (page 10, line 379).
Comment 4. The text on PIP does not belong in the perspectives, it should be in the immune system part or in the breast cancer specific immunity part.
Response: The authors agree this point and as suggested, we have moved the text on PIP to the breast cancer specific immunity section (page 8, line 315).
Comment 5. In perspectives I would prefer indications how to decide which therapy to use, or which information will be needed to collect before we can make decisions on personalized treatment in breast cancer.
Response: Again, we agree with the reviewer and as suggested, we have added a short text to this section on criteria to be considered for personalised breast cancer treatment with more focus on immunotherapy (page 12, line 447).
Reviewer 2 Report
Many thanks for this good review on Regulation of Immunity during Breast Cancer. Nevertheless, I have some considerations:
- As the authors mentioned in the Introduction section, breast cancers are a very heterogeneous group of tumors. However, I am not sure to consider invasive lobular carcinoma and invasive ductal carcinomas depending on whether the cancer affects the lobules or ducts of the breast. Indeed, the two terms were introduced a few decades ago, believing that the first form derived from the main ducts and the second from the lobules. In reality, most carcinomas arise in the terminal duct lobular unit (TDLU) and subsequently, due to mechanisms that are still not well known, give rise to different tumors, not only for their morphology but also for their biological behavior. In particular, the ductal aspect is secondary to a process of unrolling the normal lobular architecture.
- On chapter “ROLE OF THE INNATE IMMUNE SYSTEM IN BREAST CANCER IMMUNITY”, I recommend a paragraph about "EFFECTOR MOLECULES OF INNATE IMMUNITY": for example on the role of the complement system or collectin proteins, as SP-D and SP-A.
Author Response
We thank the reviewers for their constructive and insightful comments which have greatly enhanced the quality of our manuscript. We have addressed all the issues raised by the reviewers and have done additional literature search to strengthen the overall quality of the manuscript.
REVIEWER 2
Comment 1. As the authors mentioned in the Introduction section, breast cancers are a very heterogeneous group of tumors. However, I am not sure to consider invasive lobular carcinoma and invasive ductal carcinomas depending on whether the cancer affects the lobules or ducts of the breast. Indeed, the two terms were introduced a few decades ago, believing that the first form derived from the main ducts and the second from the lobules. In reality, most carcinomas arise in the terminal duct lobular unit (TDLU) and subsequently, due to mechanisms that are still not well known, give rise to different tumors, not only for their morphology but also for their biological behavior. In particular, the ductal aspect is secondary to a process of unrolling the normal lobular architecture.
Response: The authors agree with the reviewer and have modified the relevant sentences in this section of the introduction to more accurately reflect the classifications (page 1, line 33).
Comment 2. On chapter “ROLE OF THE INNATE IMMUNE SYSTEM IN BREAST CANCER IMMUNITY”, I recommend a paragraph about "EFFECTOR MOLECULES OF INNATE IMMUNITY": for example, on the role of the complement system or collectin proteins, as SP-D and SP-A.
Response: We thank this reviewer for this insightful comment and as recommended, we have now added a paragraph on “effector molecules in the innate immune system” in the introduction section (page 6, line 242).
Reviewer 3 Report
The authors have done a very decent job in giving an overall review of the big picture of the field. The manuscript is well-organized.
However, it seems that it lacks in-depth analyses, and thus a bit shallow at times. This could be overcome in a revision. One solution to this is that authors select a few most important studies (publications) and evaluate them with more critical eyes.
Two minor issues:
1. The title: the current title “regulation of immunity during breast cancer” sounds like an incomplete phrase. It might be changed to, (1). “Regulation of immunity during breast cancer progression and immunotherapy”; or (2). “Regulation of immunity in breast cancer”.
2. A number of reference need updated information:
Ref #26 The correct article number, (not page numbers), is 4273943.
Ref #40: the article number is E7.
Ref #61: article number?
Ref #72: volume and page numbers
Ref #73: ?
Ref #96: ?
Ref #103: ?
Author Response
We thank the reviewers for their constructive and insightful comments which have greatly enhanced the quality of our manuscript. We have addressed all the issues raised by the reviewers and have done additional literature search to strengthen the overall quality of the manuscript.
REVIEWER 3
MAJOR COMMENT
Comment: However, it seems that it lacks in-depth analyses, and thus a bit shallow at times. This could be overcome in a revision. One solution to this is that authors select a few most important studies (publications) and evaluate them with more critical eyes.
Response: The primary objective of this review was to evaluate the key factors that regulate immunity to Breast cancer with special emphasis on the role of prolactin inducible protein in breast tumor immunity. As a result, we feel that we did an in-depth review on the role of PIP in breast cancer. However, we do agree with the recommendation by the reviewer and have selected a few studies for more in-depth analysis. Specifically, we have expanded on studies involving the use of immunotherapeutic strategies for management of breast cancer.
MINOR COMMENTS
Comment 1: The title: the current title “regulation of immunity during breast cancer” sounds like an incomplete phrase. It might be changed to, (1). “Regulation of immunity during breast cancer progression and immunotherapy”; or (2). “Regulation of immunity in breast cancer”.
Response: We have now changed the title of the manuscript to “Regulation of Immunity in Breast Cancer”
Comment 2: A number of references need updated information:
Ref #26 The correct article number, (not page numbers), is 4273943. (now #27)
Ref #40: the article number is E7. (now #41)
Ref #61: article number? (now #62)
Ref #72: volume and page numbers (now #73)
Ref #73: ? (now #78)
Ref #96: ? (now #115)
Ref #103: ? (now #92)
Response: We are sorry for this omission and sincerely thank this reviewer for pointing this out. We have now carefully cross-checked and updated the references in the revised manuscript.
Round 2
Reviewer 2 Report
Thank you very much for editing your manuscript.
The manuscript is well written and clearly presented, although somewhat descriptive. The authors have conducted a thorough literature review, undertaken a rigorous data collection and have analyzed information accurately.
It was a pleasure to read this manuscript.
Reviewer 3 Report
The authors have improved the quality of the manuscript during revision. It is now recommended to be accepted in principle.
There is still a very minor issue regarding two references in which the authors did not make all of the right corrections during revision. However, this correction could be made during the Author Proof stage.
1. Ref #27. The page numbers (1-13) are not for citation purpose and thus should be deleted.
2. Ref #41. The authors somehow made a new mistake. The volume is 4 and the article number is E7, not the other way around. Thus, it should be, [Biomedicines, 2016, 4, E7.]